# Research on New Energy Automobile Manufacturing Service Derivatization Based on TRIZ

**Hui Wan [1], Jens Mathis Rieckmann [2,\*], Qianqian Zhang [1]**  **and Qiao Ping [1]**

[1]    Beijing Institute of Technology, Beijing 100811, China; 3120150678@bit.edu.cn (H.W.);
       candyqzh@163.com (Q.Z.); ping_qiao@163.com (Q.P.)
[2]    Fraunhofer IPK, 10587 Berlin, Germany
\*     Correspondence: mathis.rieckmann@ipk.fraunhofer.de

**Abstract:** The development of the new energy automobile industry is crucial to the industrial structure upgrading of the manufacturing industry in developing countries. The more efficient service derivation of new energy vehicles needs to be considered from the perspective of manufacturing innovation. This article discusses the feasibility of applying the theory of Teoriya Resheniya Izobretatelskikh Zadatch (TRIZ) to the service derivative of new energy vehicles and forms a preliminary idea according to the characteristics of the service derivative of the manufacturing industry. By using the analysis tool and solution of TRIZ, this paper investigates the practical problem in developing the new energy vehicle market to verify a solution. The research shows that TRIZ can effectively generate new energy automobile manufacturing service derivative schemes and explore the service derivative path.

**Keywords:** TRIZ; new energy vehicles; manufacturing service derivative; innovation path

---

## 1. Introduction

The rising global awareness of environmental protection, continuous increasing energy costs, and shortage of energy supply draw more and more attention to the development and industrialization of new energy vehicles [1,2]. The development of new energy vehicles is the key for developing countries to solve the problem of energy shortage and insufficient capacity of energy storage [3,4], and it is the urgent need for each country's energy security and industrial strategy.

Compared with developed countries, the research and development of new energy automobile manufacturing industries in developing countries fail to integrate the superior resources of disciplines. Manufacturers in developing economies also have problems in compensating for the lack of key components and core technologies. In the process of international cooperation and competition, it is necessary to enrich the understanding of the new energy automobile industry and to use service derived as a value-added method of new energy vehicles to be competitive in the absence of core technology. This requires effective integration and innovative application of existing resources.

Furthermore, there is the requirement to create greater customer value and give full play to the advantages of the automobile industry and the system integration of resources [5,6]. Therefore, the new energy automobile manufacturing industry in developing countries needs to participate in the international market competition with the dual role of manufacturer and service provider. The best practice of IBM, Rolls-Royce, and other companies shows that enterprises with emphasis on physical manufacturing have gradually shifted their business to both ends of the value chain, from product manufacturers to integrated solution providers [7]. The increasing number of manufacturing enterprises gradually integrate manufacturing business and service business, actively seek the enhancement of interaction between the two, explore innovation opportunities, and form a service innovation-driven

development mode with more high-level service, diversified products, systematic functions, and strong interaction between enterprises and customers.

From the present situation of new energy automobile manufacturing in developing countries and the developing trend of the future, new energy vehicles' mature technology still needs a long development time. Wait-and-see and follow the imitation innovation strategy of the enterprise should seize this strategic period and explore new energy automobile service value, i.e., develop new energy vehicles' potential domestic market. Service innovation based on the new energy manufacturing industry can not only release the existing productivity of enterprises but also enrich the service connotation of enterprises and lay a foundation for winning customers in the future.

Therefore, the problem of how to realize service innovation in the new energy automobile manufacturing industry in developing countries and the specific way to apply it appears and needs to be solved. Therefore, this study uses the Teoriya Resheniya Izobretatelskikh Zadatch (TRIZ) to describe the service derivation process of the new energy automobile manufacturing industry suitableness for developing countries, to provide theoretical guidance for service derivation of the new energy automobile manufacturing industry.

## 2. Theory Background

In recent years, manufacturing and service have become increasingly integrated and dependent on each other, and their boundaries have become increasingly blurred [8]. To win new customers and expand market share, the traditional manufacturing industry needs to overcome the problem of product homogeneity while technological upgrading reduces costs. After comprehensive consideration of production capital, economic benefit, social benefit, and increasingly sensitive ecological benefit, manufacturing enterprises often choose to provide more product-based nonmaterialized services [9] instead of providing specialized services for stripping products to meet customer needs. The development of new energy vehicles in developing countries is subject to the lack of core technologies and insufficient market demand. As the most profitable field of the automobile industry is the field of automobile service [10,11], the service of new energy automobiles will surely become the field of a high return in the future [12]. The development of new energy automobile service derivatives can not only lay the foundation for future market sales but also get close to consumers and closely grasp the development trend of new energy automobile products.

According to the degree of embedding, Gebauer divided service into five types, like service innovation to start a new service business (1), offer new products to current markets (2), expand existing services (3), integrate product lines (4), and change product functions (5) [13]. Some researchers regard service innovation as an integral part of the manufacturing enterprise's three stages of providing goods, additional services, and product-service packages [14,15]. Many researchers believe that service innovation involves the enterprise adding new services, expanding the original services, and improving the original service model. That means that the innovation of service business is carried out by enterprises according to customer requirements [16]. Therefore, academic circles put forward the concept of service derivative, which refers to the new service phenomenon that the manufacturing enterprises give birth to or bind to physical products by relying on the manufacturing process of physical products in a specific way [17,18].

The three concepts of service innovation, service production, and service derivation have different focuses, but their connotations all clearly express the necessity of the transformation of the manufacturing industry. On this basis, a large number of theories and case studies demonstrate the importance and feasibility of the transformation of traditional manufacturing services for economic development, especially for technological innovation and industrial upgrade [19,20]. As the research subject is the new energy automobile manufacturing industry in developing countries, the concept of service derivation emphasizing service value-added by relying on products was adopted. The academic community has reached a consensus that providing solutions instead of pure products is an important means of upgrading the manufacturing industry. However, there is no clear conclusion as to how the

new energy vehicle manufacturing industry in developing countries can achieve transformation and upgrading through service derivation. This paper will address this important research gap.

The theory of TRIZ was founded by Altshuller, a Soviet scientist [21,22]. It was first applied in technical system innovation and later expanded to nonengineering fields such as business, enterprise management, quality management, finance, education, and operation management [23]. Because the theory is abstract based on a large number of studies in patenting, its theoretical methods often span different fields and look at problems from the perspective of development, stimulating innovative thinking. The service derivation of the new energy automobile manufacturing industry needs to be based on market reality and creative theory. Applying TRIZ to explore new energy automobile manufacturing service derivative can reduce the cost of enterprise innovation transformation and expand the application of the theory.

To sum up, although there have been a lot of research results on new energy vehicle manufacturing in relevant domestic and international literature, technological breakthroughs are often taken as the development direction of the industry, and the research on its service-oriented manufacturing is still young. Research on the practical possibilities, derivation paths, and specific ways of new energy automobile manufacturing service derivation is rare [24–26]. Based on TRIZ, this study discusses the service derivation of the new energy automobile manufacturing industry and tries to fill this gap.

## 3. Applying Results

### 3.1. Feasibility Analysis

The service derivation of the new energy automobile manufacturing industry requires the integration of resources, systematic analysis of the derivation path, and implementation of specific operations. Therefore, its guiding theory needs to have a cross-field vision, a high level of systematic thinking, and a down-to-earth approach. TRIZ aims to analyze the specific requirements of the users, to identify the conflicts in the implementation difficulties of the specification, to describe the ultimate ideal solution to solve the problem, and finally to apply the solution to the specific problem. The application of TRIZ to new energy automobile manufacturing service derivation has the following characteristics:

1. Service derivative is a process in which a manufacturing enterprise transforms or expands its original business. Its purpose is to finally realize the transformation and upgrading of the traditional manufacturing business and the co-creation of value between the supply and demand side. Therefore, decision-making in the process of service derivation requires systematic and collaborative thinking. Once the service becomes the profit source of the manufacturing enterprise, the guarantee that the customer timely and accurately participates in the service customization builds the source of the enterprise to provide the service. Although the Kano model and the multilevel deductive analysis method of quality function development can also absorb customer opinions to form decisions, they are not based on a business value creation. In the process of applying TRIZ, conflicts in the operation process can be analyzed and solved, and service derivative innovation can be carried out from the perspective of manufacturing transformation.

2. Service derivative requires manufacturing enterprises to provide services or develop potential services according to customer needs. Its fundamental purpose is to form unique competitiveness in the market competition and finally achieve a high rate of return to obtain profits. Therefore, there must be a contradiction between short-term gain and long-term return, between self-interest and serving customers, technological breakthroughs, and service improvement. For the resolution of conflicts, the traditional solution is to obtain the maximum benefit or bear the minimum loss through a "compromise plan", while TRIZ adjusts the parameters of contradictions by analyzing the attributes of contradictions and finally resolves the contradictions. In the process of service derivation in the new energy manufacturing industry, its ability to solve contradictions is also the embodiment of service value in essence.

3.  Service derivative of manufacturing enterprises needs to integrate different types of resources, which is also the key to the value creation of service derivative. Therefore, how to coordinate resources systematically becomes an important issue derived from the service of the new energy automobile manufacturing industry. TRIZ is guided by dialectics, system theory, and epistemology. It takes the technical system evolution method as the theoretical backbone; takes the technical process, the contradiction, resources, and the ideal final result (IFR) as a basic concept; and takes the analysis tool, the solution tool, and the problem-solving process as the operation tool. TRIZ can not only focus on customer needs but also coordinate resources to resolve conflicts and form the optimal service derivative implementation scheme.

### 3.2. Basic Tools

Operators of TRIZ have analysis tools, solution tools, and problem-solving processes. The solution starts with defining problems and finding the initial ideas to solve problems by describing the working principles, main problems, and occurrence conditions of things. This stage for the application of TRIZ is defined as project description.

After the specification describes the problem to be solved, the system analysis function model combines the causal relationship of the problem decomposition to begin finding a solution. In the TRIZ solving stage, according to the different solutions, one or more solution methods can be applied to obtain a problem's solution, and then the proposed solution can be further processed, including storing the innovation experience and results obtained from this solution in the relevant database. Its framework and process are shown in Figure 1.

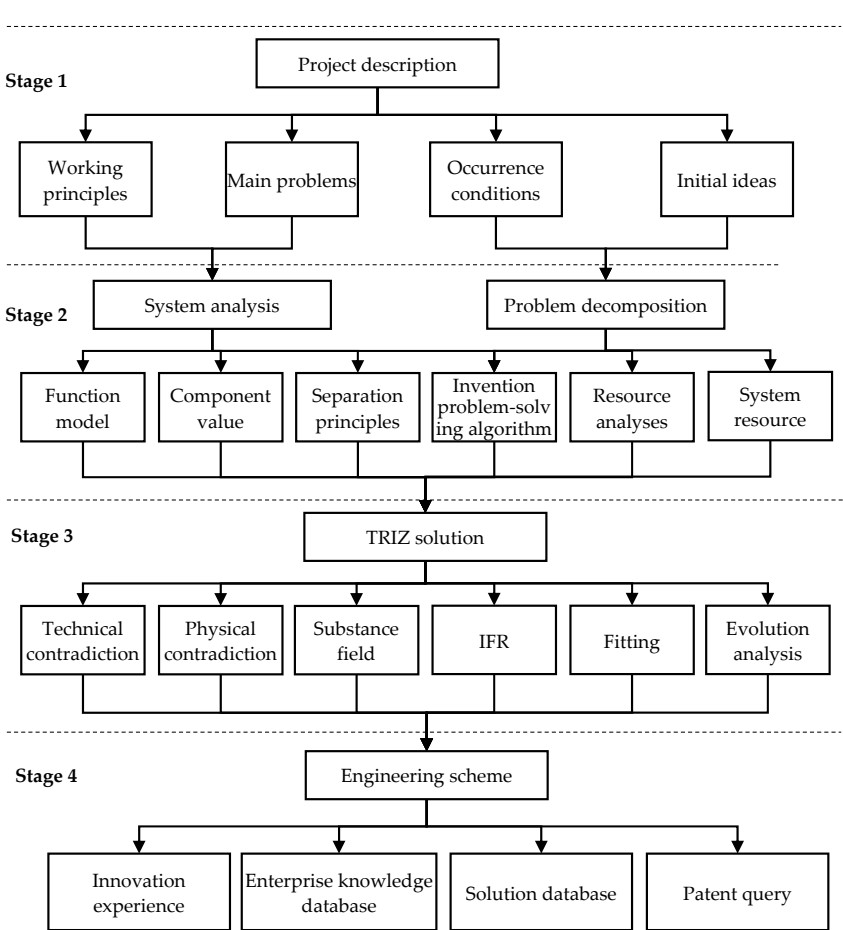

**Figure 1.** The framework of TRIZ tools. TRIZ: Teoriya Resheniya Izobretatelskikh Zadatch; IFR: ideal final result.

TRIZ applies to the new energy automobile manufacturing industry scientific and effective service derivation methods. This includes mainly functional analysis, idealization process, resource analysis, conflict analysis, and the generation of innovative solutions.

### 3.2.1. Functional Analysis

By establishing the functional model, the system components and their functions can be analyzed more intuitively. From the perspective of the product function generation, the idea of value engineering can be used to analyze the system, subsystem, and parts that constitute the product.

### 3.2.2. Ideality

The application of TRIZ to the solution is dynamic, and the process from the initial innovative solution to the ultimate IFR is a process of constantly optimizing its feasibility and efficiency. The parameter to measure the feasibility and efficiency of the solution is the ideal level.

### 3.2.3. Resource Analysis

Since the realization of the function requires the support of resources, it is necessary to fully consider the resources in the solution process, such as "what resources have been owned", "what resources can be obtained", "how to use the resources", "how to deal with the consequences", etc. In this case, resources are broadly defined and can be classified according to their environment, utilization, and physical properties, as shown in Figure 2.

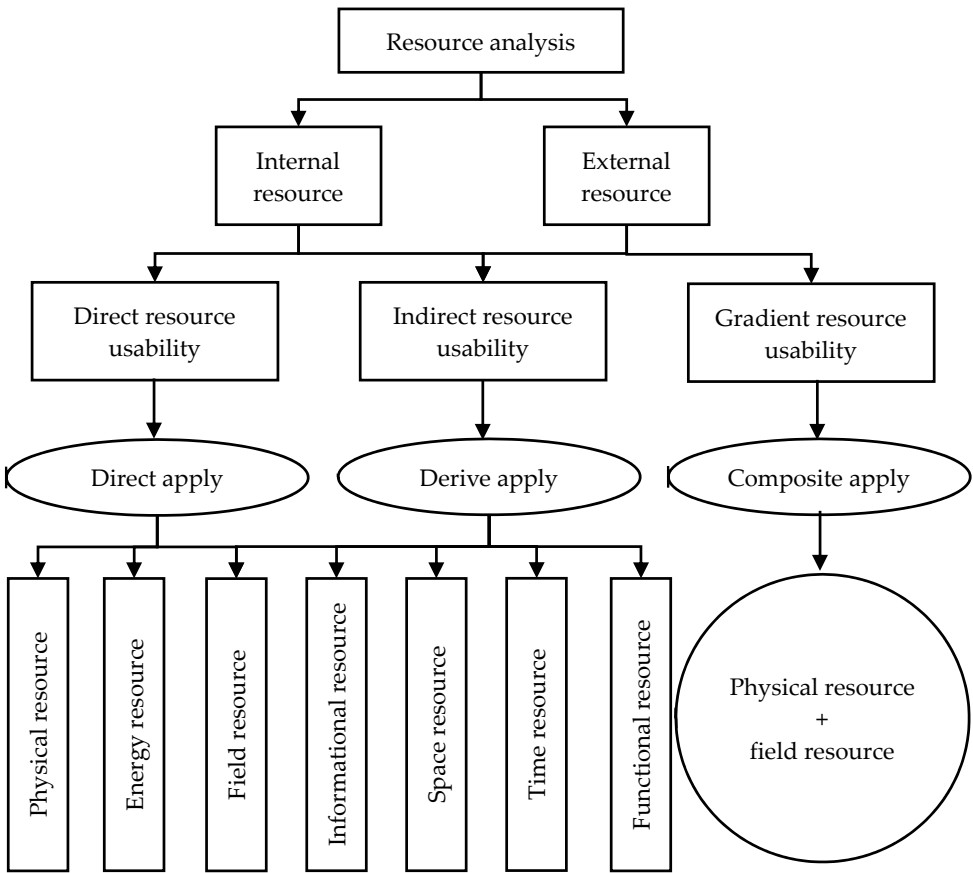

**Figure 2.** Classification of resources in TRIZ.

### 3.2.4. Contradiction

A conflict is inevitable in the process of service innovation. An example is improving strength by adding more material. This solution often makes weight get worse. The useful function (UF) is the benefit provided by the system, and harmful function (HF) is the unwanted output of the system. In the example above, strength presents the UF parameter and weight is the HF parameter. Contradictions are indicative of inventive problems arising from the apparent incompatibility of desired features within a system, which can be divided into physical contradiction and technical contradiction. The technical contradiction refers to the contradiction between two common parameters, and the physical contradiction refers to the demand contradiction which is mutually exclusive for the same parameter, such as a knife which has a sharp part to cut and a blunt part to hold. Both conflicts can be formatted in the TRIZ, as shown in Figure 3:

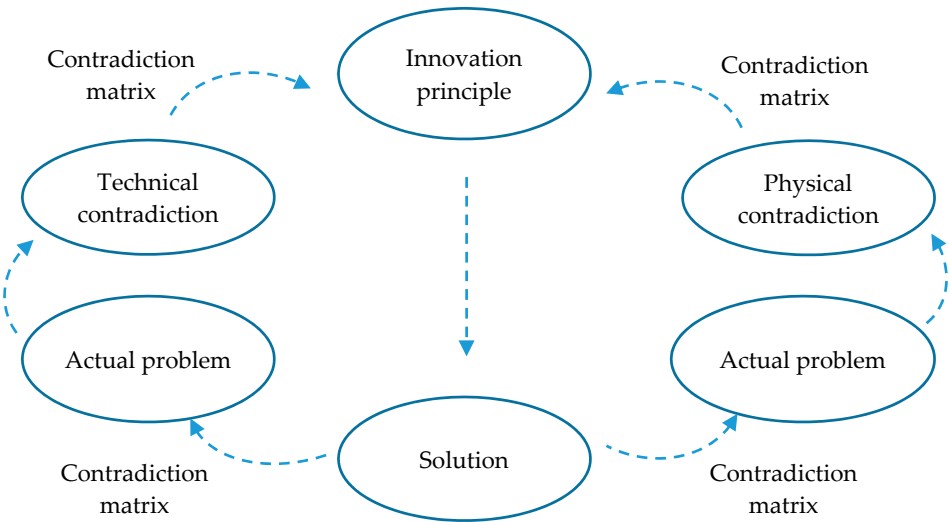

**Figure 3.** The conflict resolution model in TRIZ.

Among them, the solution of technical contradiction depends on 39 technical parameters and a contradiction matrix, while the solution of physical contradiction is realized by separating the characteristics of contradiction. If the solution is still not ideal, then is it necessary to continue to subdivide the contradiction in the actual problem, to find a solution.

### 3.2.5. Innovational Solution

If an innovative solution is generated in the analysis phase, the scheme is moved to the next ideal level comparison link. In the process of solving TRIZ, the solution method is selected according to the characteristics of the problem, to obtain innovative solutions. It is not necessary to use all methods. After the solution is completed, it is taken as the standard to select the scheme and put it into practice.

### 3.3. Results

Through the above analysis, it is shown that new energy vehicle service derivatives need innovative ideas in design and solutions. TRIZ provides the required systematic principles, a toolkit consisting of methods, which can support the process in at least two aspects:

1.  The problem can be matched with one or more of the conceptual solutions, such as the five previously mentioned methods. These identified conceptual solutions can afterward be transformed into a specific, factual solution that provides an answer to the original factual problem.

2. Different from the conceptual analysis of the theoretical model, the application of TRIZ to the service derivation of new energy vehicles can effectively promote the evolution of new energy vehicle technologies and service systems.

Therefore, this approach is trying to find specific factual solutions to factual problems directly. At the same time, this paper also wants to test the applicability of TRIZ through the practical difficulties derived from new energy vehicle services. This is a reciprocal approach in which theory guides practice and practice feeds back theory.

## 4. Discussion

### 4.1. System-Verb-of-Projects (SVOP) to Derive the Conceptual Problem

An important feature of TRIZ is the attempt to establish a formatted method that can solve innovative problems, so special emphasis is regularly placed on the description and analysis of problems before solving them. To standardize seemingly complex, life-like, or specialized problems, SVOP is applied to describe that the target system needs to change the parameters of the target object, for example, "Increase (verb) the customers' demand (project) in the field of new energy automobile manufacturing (system)", where the demand degree is the parameter of the system acting object. Based on the analysis of the general needs of target customers and the characteristics of new energy vehicles, the explicit and implicit needs of current customer groups are analyzed.

New energy vehicles often need to be charged, which makes it difficult to drive long distances, resulting in low purchase willingness of consumers. Under the premise that the new energy battery technology is difficult to break through, this problem can be split into existing smallest problems. For example, "new energy vehicles cost a long time to travel" and "consumers are not willing to buy new energy vehicles". Then, solve the unit problem one by one.

### 4.2. Ideal Final Result (IFR)

The purpose of functional analysis in TRIZ is to analyze systems, subsystems, and components from the perspective of completing functions rather than from the perspective of technology. Through functional analysis, cost and complexity can be reduced. The IFR of the problem is obtained when the useful function of the solution is the largest and the harmful function is the smallest. Although different fields have different preferences and different evaluation criteria, IFR has the following characteristics:

- It maintains the advantages of the original system.
- It eliminates the deficiency of the original system.
- It does not make the original system more complex.
- No new defects are introduced.

Overall, low cost, high performance, high reliability, no pollution, low consumption is the ideal state for most products or solutions.

Therefore, IFR derives from the service of new energy automobile manufacturing enterprises. It is the most abundant service business with new energy automobiles as the carrier. The original function of the new energy automobile manufacturing industry is not weakened to enhance the service function. While maintaining the normal operation of the original manufacturing industry with a wider business scope, it can meet the implicit and explicit needs of customers without increasing the complexity of the enterprise organizational structure.

### 4.3. Contradiction Matrix

To solve the technical contradictions in TRIZ, the specific problems to be solved described in popular language should be transformed into the technical contradictions described by using 39 common engineering parameters. According to the problem types, the contradiction matrix

should be used to find the corresponding innovation principles, and then the specific solutions to the contradictions can be found according to the principles. Before breakthroughs were made in core technologies, new energy vehicle sales were largely limited by the "high cost of long-distance driving time". The conflict in this dilemma is that the more times a new energy car is charged, the farther it will travel, but the longer the delay. Once the time parameter "reducing charging time" is optimized, the "driving distance" parameter is worsened. That displays that there is a technical contradiction between the two parameters of "time loss" when charging new energy vehicles and "energy consumption of moving objects". By looking at the contradiction matrix table, the innovation principles 18, 19, 35, and 38 are obtained, as shown in Figure 4:

Improved parameter: 25 Loss of time

Deteriorating parameter: 19 Use of energy by moving object

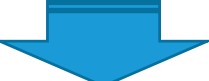

Solution:      35 Adaptability or versatility

38 Extent of automation

19 Use of energy by moving object

18 Illumination intensity

**Figure 4.** Use of the contradiction matrix.

Using principle 19 of innovation—periodic actions (continuous action is changed by adding periodic-based action)—the following solutions can be obtained:

- Provide official vehicle rental service for the government; design shuttle buses for economic development zone; offer special new energy vehicles for municipal administration, public security, traffic management, environmental protection, safety, and other fields; and carry out special rental cooperation for large hotels, tourist attractions, exhibitions, airports, and other consumer groups with strong consumption capacity;
- Develop intra-city and inter-city logistics leasing services for e-commerce customers;
- Carry out the rental of passenger cars for short-distance commuting; and develop programs for differentiated markets to meet individual consumer demands for car rental, e.g., in large communities, business districts, and industrial parks with dense residential populations, as well as commuting or business people with fixed routes, long-term or time-sharing rental service requirements;
- Offer event-based service, like regional scenic spots rental cars and wedding car rental services.

### 4.4. Substance-Field Model (Su-F Model)

The Su-F model analysis is an important description and analysis method in TRIZ. It starts from the interaction between substances and makes a comprehensive analysis of the interaction mode or mechanism between substances, it includes the field and their impact on the required functions of the system, to clarify the possible methods and directions of solving problems. Altshuller believed that any existing function is composed of the receiver ($S_1$), actuator ($S_2$), and field (F) [26], and "substance-field" is the smallest controllable technical model with working capacity. By building the

substance-field model, the interaction between the components in the problem to be solved can be accurately described. The corresponding basic solution can be found according to the type of model. For example, the problem of "car buyers are not willing to choose new energy vehicles" mentioned above can be analyzed by building a substance-field model, as shown in Figure 5.

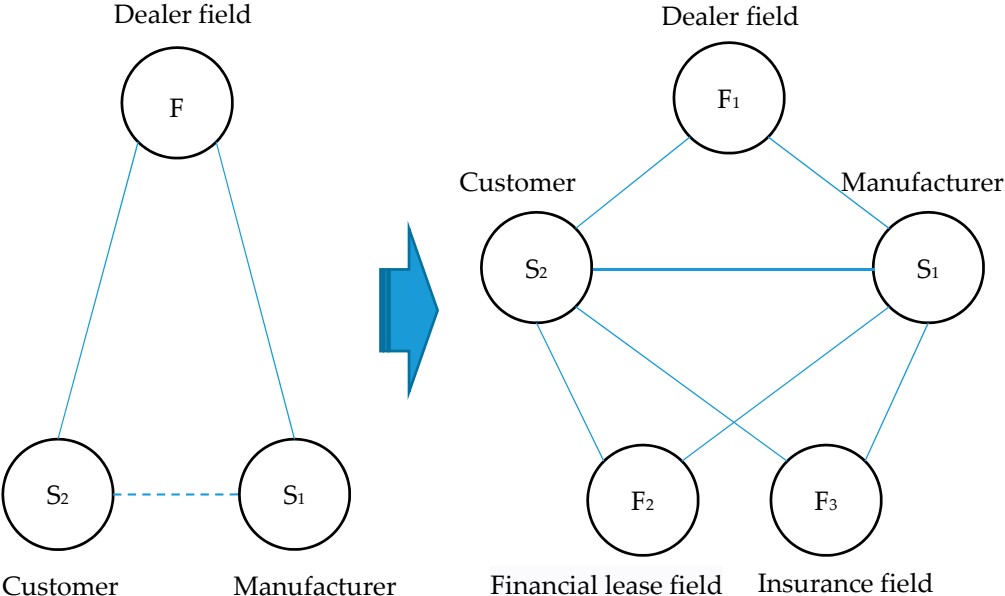

**Figure 5.** Su-F model of new energy vehicle service derivative.

In the traditional Su-F model of the automobile trade, the field between buyer ($S_2$) and manufacturer ($S_1$) is only the automobile trading field (F). The practical difficulty is that buyers are not willing to choose new energy vehicles. That means, the effect of $S_2$ to $S_1$ is insufficient. In TRIZ, 23 standard solutions on the second level of the standard method and six standard solutions on the third level of the standard method can be applied to solve the problem of completing the matter field model with insufficient effect. Selecting the standard solution 15, dual matter field model, the useful function f of the existing system is insufficient and needs to be improved, but new elements and substances are not allowed to be introduced. In this case, a second field $F_2$ can be added to enhance the effect of $F_1$. As buyers worry about new energy vehicles, they mainly focus on battery life technology and insurance claims. Therefore, the introduction of the financial leasing market ($F_2$) and the insurance agency trading market ($F_3$) are considered. By introducing $F_2$ and $F_3$, more relationships between the new energy vehicle manufacturing industry and consumers can be introduced in order to develop new energy purchase loan agencies, insurance, and other services.

*4.5. Ideality*

The invention problem-solving algorithm ARIZ (Algoritm Resheniya Izobretatelskikh Zadatch) in TRIZ is a solution method in the process of invention problem-solving. The solution method usually starts from the smallest problem that can be decomposed, and continuously supplements and improves the formed scheme, finally approaching the ideal solution of the overall scheme. However, the idealization level of different plans is not identical, namely, "although each is the best, the comparison is better".

The ideality is a concept of an ideal system that is virtual, provides the required functions, and produces no undesirable side effects. The ideal principle is increasing the ratio of useful effects to harmful effects.

$$I = \sum UF \div \left( \sum HF + \sum Cost \right) \tag{1}$$

As the factors involved in the service derivative of the new energy automobile manufacturing industry include not only the production capacity related to new energy automobile manufacturing but also its ability to provide services, it is necessary to comprehensively analyze $\sum$UF, $\sum$HF, and $\sum$Cost of innovative solutions.

The overall scheme with the highest idealized level (I) is selected to make the service derivation of new energy manufacturing enterprises more practical and scientific.

## 5. Conclusions

The paper takes advantage of the TRIZ cross-field innovation to solve the development dilemma of the lack of core technology in the new energy vehicle manufacturing industry and the difficulty of market development. Through the feasibility analysis of the service derivation of the new energy automobile manufacturing industry, the paper proposes a basic solution. Corresponding TRIZ analysis tools and solutions according to the characteristics of manufacturing service derivatives were selected. Finally, the thesis provides a derivative solution for the service of the new energy vehicle manufacturing industry.

The research results show that the TRIZ theoretical innovation method can effectively put forward the new energy automobile manufacturing industry service derivative program, explore the service innovation path, help the new energy automobile manufacturing industry diversification of business mode, and complete the industrial structure upgrades.

The conclusion of this study provides enlightenment from the theoretical, governmental, and enterprise level as to how to effectively promote the application of TRIZ in service derivative of the new energy automobile manufacturing industry:

- Theoretical research level: TRIZ should continue to expand and deepen the research of nontechnical dimension TRIZ with cross-field and low entropy as the research direction, to promote TRIZ to play a greater theoretical supporting role in innovative practice in new fields.
- Level of government industrial strategy: Government departments should organize trainings, exchanges, and symposiums on TRIZ, encourage colleges and universities to offer relevant courses, and actively meet the needs of new energy automobile manufacturing enterprises for service derivative cooperation to seek win–win results through cooperation.
- Operation of new energy automobile manufacturing enterprises: New energy automobile manufacturing enterprises should actively seek service opportunities and paths, formulate effective incentive mechanisms to encourage the learning of TRIZ, and take the innovation theory of TRIZ as an important content of enterprise development strategy.

**Author Contributions:** Conceptualization, H.W. and Q.Z.; methodology, H.W.; software, H.W.; validation, H.W., Q.Z., Q.P., and J.M.R.; formal analysis, H.W.; investigation, H.W. and J.M.R.; resources, H.W., J.M.R.; data curation, H.W.; writing—original draft preparation, H.W.; writing—review and editing, J.M.R.; visualization, J.M.R.; supervision, H.W.; project administration, H.W.; funding acquisition, H.W. All authors have read and agreed to the published version of the manuscript.

**Funding:** This research was funded by the MINISTRY OF SCIENCE AND TECHNOLOGY INNOVATION METHOD WORK SPECIAL ISSUE, grant number 2015IM030100.

**Conflicts of Interest:** The funders had no role in the design of the study; in the collection, analyses, or interpretation of data; in the writing of the manuscript, or in the decision to publish the results.

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
