# Peer review of "Research on New Energy Automobile Manufacturing Service Derivatization Based on TRIZ"

_sustainability, doi:10.3390/su12166652_

Round 1

Reviewer 1 Report

It was unclear as to whether the paper was to present the solution or the application of TRIZ to the problem. It would be useful to illustrate the use of the TRIZ process in the development of one or more new energy automobile service schemes. I do not believe the problem was stated clearly enough to be able to follow it all the way through the analysis. The contradiction matrix was very clear, but different considerations appear to be used in the SuField study.

In section 3 it is not clear what the results were. Need to concisely state what the TRIZ has provided you otherwise it is just a discussion of what TRIZ is, not how it was applied.

Please fix and resubmit - this is important work. Perhaps just presenting one of the possibilities and tracing through the methods to show how that possibility was arrived at might make it a clearer read.

Line 107-8 Can you provide one or more references? Or were you unable to find any at all? 

Author Response

Response to Reviewer 1

Point 1: It was unclear as to whether the paper was to present the solution or the application of TRIZ to the problem. It would be useful to illustrate the use of the TRIZ process in the development of one or more new energy automobile service schemes. I do not believe the problem was stated clearly enough to be able to follow it all the way through the analysis. The contradiction matrix was very clear, but different considerations appear to be used in the SuField study.

Response 1: Thank you for your feedback. Our original writing does cause ambiguity. We revised it according to the reviewer's comment as follows:

1) Emphasize in the abstract that:

By using the analysis tool and solution of TRIZ, this paper investigates the practical problem in developing the new energy vehicle market to verify a solution

2) State in part 3.3 (results) that:

This is a reciprocal approach in which theory guides practice and practice feeds back theory.

3) Illustrate the Su-F model in adding new fields to solve insufficient effect:

In the traditional Su-F model of the automobile trade, the field between buyer (S1) and manufacturer (S2) is only the automobile trading field (F). The practical difficulty is that buyers are not willing to choose new energy vehicles. That means, the effect of S2 to S1 is insufficient. In TRIZ, 23 standard solutions on the second level of the standard method and six standard solutions on the third level of the standard method can be applied to solve the problem of completing the matter field model with insufficient effect. Selecting the standard solution 15, dual matter field model, the useful function f of the existing system is insufficient and needs to be improved, but new elements and substances are not allowed to be introduced. In this case, a second field F2 can be added to enhance the effect of F1. As buyers worry about new energy vehicles, they mainly focus on battery life technology and insurance claims. Therefore, the introduction of the financial leasing market (F2) and the insurance agency trading market (F3) are considered. By introducing F2 and F3, more relationships between the new energy vehicle manufacturing industry and consumers can be introduced in order to develop new energy purchase loan agencies, insurance, and other services.

Point 2: In section 3 it is not clear what the results were. Need to concisely state what the TRIZ has provided you otherwise it is just a discussion of what TRIZ is, not how it was applied.

Response 2: Thank you for your advice. This part is inadequate. We add the part 3.3 Results to state the feasible ideas and specific methods guidance for new energy vehicle service derivative issues through TRIZ in two parts as followed:

Through the above analysis, it is shown that new energy vehicle service derivatives need innovative ideas in design and solutions. TRIZ provides the required systematic principles, a toolkit consisting of methods, which can support the process in at least two aspects:

  1. The problem can be matched with one or more of the conceptual solutions, such as the five previous mentioned methods. These identified conceptual solutions can afterward be transformed into a specific, factual solution that provides an answer to the original factual problem.
  2. Different from the conceptual analysis of the theoretical model, the application of TRIZ to the service derivation of new energy vehicles can effectively promote the evolution of new energy vehicle technologies and service systems.

Point 3: Please fix and resubmit - this is important work. Perhaps just presenting one of the possibilities and tracing through the methods to show how that possibility was arrived at might make it a clearer read.

Response 3: Thank you for your suggestion. We have adopted the revised model in MS and hope to improve the readability.

Point 4: Line 107-8 Can you provide one or more references? Or were you unable to find any at all?

Response 4: In fact, in the writing of the original manuscript, we have searched the relevant literature carefully, but we have obtained very little. It can be said that “new energy vehicle”+“manufacturing service” as keywords in the WOS (Web of Science), cannot find any literature. We still try to find some more input from other related literature and cross-references to support the argument that TRIZ is helpful in finding practical possibilities, derivation paths, and specific ways of new energy automobile manufacturing service derivation.

Reviewer 2 Report

The article is interesting and generally, it deserves to be published with some revisions that are suggested below:

  • According to TRIZ, a field is the effect that it has on an object (substance). This effect changes or maintains the properties of the object. Between the field and the substances in the Su-Field models, one or two-way relations are specified. Could you explain the relations in your article or make a table to state it.

  • Would you please let us know why do not apply the Problem Hierarchy Analysis (PHA) of TRIZ that can solve several important points, such as business opportunities, and finding better points to tackle a problem in your article.

3) You may try to search more relating References between 2018 to 2019.

Author Response

Response to Reviewer 2

Point 1: The article is interesting and generally, it deserves to be published with some revisions that are suggested below:

According to TRIZ, a field is the effect that it has on an object (substance). This effect changes or maintains the properties of the object. Between the field and the substances in the Su-Field models, one or two-way relations are specified. Could you explain the relations in your article or make a table to state it.

Response 1: Thank you for your comment and we are impressed by your understanding of TRIZ field and substance. We revised it according to the reviewer's opinion as follows in part 4.4. (Substance-field model):

3) Illustrate the Su-F model in adding new fields to solve insufficient effect:

In the traditional Su-F model of the automobile trade, the field between buyer (S1) and manufacturer (S2) is only the automobile trading field (F). The practical difficulty is that buyers are not willing to choose new energy vehicles. That means, the effect of S2 to S1 is insufficient. In TRIZ, 23 standard solutions on the second level of the standard method and six standard solutions on the third level of the standard method can be applied to solve the problem of completing the matter field model with insufficient effect. Selecting the standard solution 15, dual matter field model, the useful function f of the existing system is insufficient and needs to be improved, but new elements and substances are not allowed to be introduced. In this case, a second field F2 can be added to enhance the effect of F1. As buyers worry about new energy vehicles, they mainly focus on battery life technology and insurance claims. Therefore, the introduction of the financial leasing market (F2) and the insurance agency trading market (F3) are considered. By introducing F2 and F3, more relationships between the new energy vehicle manufacturing industry and consumers can be introduced in order to develop new energy purchase loan agencies, insurance, and other services.

Point 2:Would you please let us know why do not apply the Problem Hierarchy Analysis (PHA) of TRIZ that can solve several important points, such as business opportunities, and finding better points to tackle a problem in your article.

Response 2: As you said, applying the PHA can solve several important points, such as business opportunities, and it is an effective method to focus on finding better points to tackle a problem. In this paper, the focus process of the problem has been completed, that is, the dilemma of service derivation of new energy vehicles, so this method is not introduced and applied in detail. Thanks once again for your comment.

Point 3: You may try to search more relating References between 2018 to 2019.

Response 3: Thank you for your advice. We have added references like:

Jiang, C.; Zhang, Y.; Zhao, Q.; Wu, C. The impact of purchase subsidy on enterprises’ R&D efforts: evidence from China’s new energy vehicle industry. Sustainability 2020, 12, 1105-1115, https://doi.org/10.3390/su12031105.

Ardito, L.; Petruzzelli, A.M.; Ghisetti, C. The impact of public research on the technological development of industry in the green energy field. Technol. Technological Forecasting and Social Change 2019, 144, 25–35, https://doi.org/10.1016/j.techfore.2019.04.007.

Ran, L.; Wu, D.; Jiao, Z.; Wang, S.; Yuan, S. Distributionally robust chance-constrained vehicle scheduling with uncertain demand. Systems Engineering-Theory & Practice 2018, 38, 1792-1801.

Xiong, Y.; Huang, T.; Li, X. Regional differences in the implementation effect of New Energy Vehicle consumption promotion policy: comparative perspectives on purchase' and use' links. China Population Resources and Environment 2019, 29, 71-78.

Gong, B.G.; Xia, X.; Cheng, J.S. Supply-Chain Pricing and Coordination for New Energy Vehicles Considering Heterogeneity in Consumers' Low Carbon Preference. Sustainability 2020, 12, 14, doi:10.3390/su12041306.

Round 2

Reviewer 1 Report

page 4 caption not with picture

Line 29 - wording  awkward

Lines 77-78, 90 - grammar errors

Lines 181-182 incorrect statement; harmful increasing degrades useful or vice versa - method requires there be an increasing factor contradicting with a decreasing factor

Lines 218-219 example has grammar error

Line 261 - should probably have made picture with principle 19 since that is what is discussed as the paper continues

Lines 312-313 - change word benefits to useful effects because of equation

Author Response

Dear Sir or Madam,

Thank you for your review. According to your suggestions, we have adjusted the article.

Point 1: Page 4 - caption not with picture

Response 1: The caption is placed under figure 1.

Point 2: Line 29 - wording awkward

Response 2: The wording and structure of the sentence have been changed like this:

Compared with developed countries, the research and development of new energy automobile manufacturing industries in developing countries fail to integrate the superior resources of disciplines. Manufacturers in developing economies have also problems to compensate the lack of key components and core technologies.

Point 3: Lines 77-78, 90 - grammar errors

Response 3: The grammar errors in three sentences have been corrected like this:

Many researchers believe that service innovation involves the enterprise adding new services, expanding the original services, and improving the original service model. That means the innovation of service business is carried out by enterprises according to customer requirements.

The academic community has reached a consensus that providing solutions instead of pure products is an important means of upgrading the manufacturing industry. However, there is no clear conclusion how the new energy vehicle manufacturing industry in developing countries can achieve transformation and upgrading through service derivation. This paper will address this important research gap.

Point 4: Lines 181-182 - incorrect statement; harmful increasing degrades useful or vice versa - method requires there be an increasing factor contradicting with a decreasing factor

Response 4: The paragraph has been rewritten and an example for the specific contradiction case has been added:

A conflict is inevitable in the process of service innovation. An example is improving strength by adding more material. This solution often makes weight get worse. The useful function (UF) is the benefit provided by the system and harmful function (HF) is the unwanted output of the system. In the example above, strength presents the UF parameter and weight is the HF parameter. Contradictions are indicative of inventive problems arising from the apparent incompatibility of desired features within a system, which can be divided into physical contradiction and technical contradiction.

Point 5: Lines 218-219 - example has grammar error

Response 5: The wording of the example have been changed like this:

An important feature of TRIZ is the attempt to establish a formatted method that can solve innovative problems, so special emphasis is regularly placed on the description and analysis of problems before solving them. To standardize seemingly complex, life-like, or specialized problems, SVOP is applied to describe that the target system needs to change the parameters of the target object, for example, “Increase (verb) the customers' demand (project) in the field of new energy automobile manufacturing (system).”, where the demand degree is the parameter of the system acting object.

Point 6: Line 261 - should probably have made picture with principle 19 since that is what is discussed as the paper continues

Response 6: An explanation is added in order to support the solutions that are presented:

Using principle 19 of innovation – periodic actions (continuous action is changed by adding periodic-based action) – the following solutions can be obtained:

  • Provide official vehicle rental service for the government; design shuttle buses for economic development zone; offer special new energy vehicles for municipal administration, public security, traffic management, environmental protection, safety, and other fields; and carry out special rental cooperation for large hotels, tourist attractions, exhibitions, airports and other consumer groups with strong consumption capacity
  • Develop intra-city and inter-city logistics leasing services for e-commerce customers
  • Carry out the rental of passenger cars for short-distance commuting; and develop programs for differentiated markets to meet individual consumer demands for car rental, e.g. in large communities, business districts and industrial parks with dense residential populations, as well as commuting or business people with fixed routes, long-term or time-sharing rental service requirements
  • Offer event-based service, like regional scenic spots rental cars and wedding car rental services

Point 7: Lines 312-313 - change word benefits to useful effects because of equation

Response 7: The word has been changed:

The ideality is a concept of an ideal system that is virtual, provides the required functions, and produces no undesirable side effects. The ideal principle is increasing the ratio of useful effects to harmful effects.
